# ZIF-8-Derived Hollow Carbon for Efficient Adsorption of Antibiotics

**DOI:** 10.3390/nano9010117

**Published:** 2019-01-18

**Authors:** Hongmei Tang, Wenyao Li, Haishun Jiang, Runjia Lin, Zhe Wang, Jianghong Wu, Guanjie He, Paul Robert Shearing, Dan John Leslie Brett

**Affiliations:** 1School of Materials Engineering, Shanghai University of Engineering Science, Shanghai 201620, China; neview2015@163.com (H.T.); jhaishun@163.com (H.J.); 18317188857@163.com (Z.W.); 2Electrochemical Innovation Lab, Department of Chemical Engineering, University College London, London WC1E 7JE, UK; runjia.lin.16@ucl.ac.uk (R.L.); p.shearing@ucl.ac.uk (P.R.S.); 3College of Health Science and Environmental Engineering, Shenzhen Technology University, Shenzhen 518118, China; wujianghong@sztu.edu.cn

**Keywords:** ZIF-8, hollow carbon, antibiotics, adsorbent

## Abstract

The harmful nature of high concentrations of antibiotics to humans and animals requires the urgent development of novel materials and techniques for their absorption. In this work, CTAB (Cetyltrimethyl Ammonium Bromide)-assisted synthesis of ZIF-8 (zeolitic imidazolate framework)-derived hollow carbon (ZHC) was designed, prepared, and used as a high-performance adsorbent, and further evaluated by Langmuir and Freundlich isothermal adsorption experiments, dynamic analysis, as well as theoretical calculation. The maximum capacities of ZHC for adsorbing tetracycline (TC), norfloxacin (NFO), and levofloxacin (OFO) are 267.3, 125.6, and 227.8 mg g^−1^, respectively, which delivers superior adsorptive performance when compared to widely studied inorganic adsorbates. The design concept of ZIF-8-derived hollow carbon material provides guidance and insights for the efficient adsorbent of environmental antibiotics.

## 1. Introduction

Due to their outstanding medical properties, antibiotics are broadly utilized to treat infectious diseases; this leads to high levels of antibiotics in hospital effluent and sewage treatment plants [1]. Since antibiotics cannot be totally metabolized by humans or animals, and it is impossible to prevent the mass utilization of antibiotics, a vicious cycle forms and results in the uncontrolled accumulation of residual antibiotics in the environment [2]. Antibiotics such as quinolones and tetracycline are able to make bacteria resistant to drugs and cause various diseases. Therefore, with the purpose of preventing the pollution of antibiotics, materials with tremendous absorption ability for antibiotics are urgently required.

In recent years, metal-organic frameworks (MOFs) have been synthesized via compounds composed of inorganic metal ions centers and organic ligands [3]. A great number of targeted applications can be achieved through the application of MOFs, owing to their diversity, as they are composed of various metal ions and organic ligands. Additionally, various morphologies of porous nanomaterials can be prepared by using MOFs as sacrificial templates through different thermal and/or chemical treatments. For instance, highly porous carbons can be produced via the heat treatment of MOFs in an inert atmosphere, coupled with chemical etching for the removal of the surface metal ions, thus, leading to an increased specific surface area. Compared with carbonaceous materials fabricated by conventional precursors, MOF-derived carbons often exhibit controllable porous architectures, pore volumes, and surface areas [4]. Various porous carbon materials, with controlled morphologies from 0 to 3 dimensions, have been successfully derived through the carbonization of MOFs, highlighting their versatility as precursors [5,6,7]. In addition to porous materials, MOF-derived carbons can be converted into hollow structures.

In this work, CTAB (Cetyltrimethyl Ammonium Bromide)-assisted synthesis of ZIF-8, a subclass of MOFs, was applied as a sacrificial template to synthesize ZIF-8 (zeolitic imidazolate framework)-derived hollow carbon (ZHC) nanostructures, with the aim of exploring highly efficient absorbent materials. The maximum adsorption capacities of ZHC for tetracycline (TC), norfloxacin (NFO), and levofloxacin (OFO) are investigated according to the data obtained from Langmuir and Freundlich isothermal adsorption simulation, as well as kinetic experiments. The adsorptive performance of ZHC is comparable to the cutting-edge inorganic adsorbents and paves a new way for the design concept of adsorbents.

## 2. Experimental

### 2.1. Chemicals

Zinc nitrate hexahydrate (Zn(NO_3_)_2_∙6H_2_O), tannic acid (TA) and cetyltrimethylammonium bromide (CTAB, analytical purity) were all purchased from Sinopharm Chemical Reagent Co., Ltd. (Shanghai, China). Anhydrous methanol (AR) and hydrochloric acid (HCl, 36%) were purchased from Shanghai Lingfeng Chemical Reagent Co., Ltd. (Shanghai, China). 2-methylimidazole (AR, 98%) was purchased from in Aladdin Industrial Corporation (Shanghai, China).

### 2.2. ZIF-8 Derived Hollow Carbon (ZHC) Preparation

Firstly, 810 mg Zn(NO_3_)_2_∙6H_2_O were dissolved in 40 mL of anhydrous methanol solution with continuous stirring, which was named Solution A. 526 mg of 2-methylimidazole were added into 40 mL of anhydrous methanol solution with continuous stirring until a homogeneous solution formed, which was named Solution B. Then, Solution A and B were mixed and a further 10 mg of CTAB was added, the mixture was sonicated for 10 min then sealed for 24 h at room temperature. The obtained white ZIF-8 products were washed with anhydrous methanol several times. After drying, the products were transferred to a 40 mL tannic acid solution (0.01 mol L^−1^), sonicated for 5 min, then collected by centrifugation and washed with water and methanol successively. After drying, the samples were put into a tube furnace and heated at 800 °C for 2 h under N_2_ atmosphere. Finally, the collected black powders were immersed in a 1 mol L^−1^ of HCl solution to remove the residual Zn or ZnO. Finally, the samples were dried under vacuum at 100 °C to obtain the final products after washing with deionized water several times. A simple schematic illustration of this process is shown in Figure 1.

### 2.3. Characterization

The microstructure of the material was characterized by scanning electron microscopy (SEM, S-4800, Hitachi, Japan). Transmission electron microscopy (TEM) analysis was performed using a JEM-2100F system (Rigaku, Japan). The X-ray diffraction (XRD) was carried out with a Cu Kα radiation source (Rigaku, Japan). The Brunauer-Emmett-Teller (BET) method was used to provide the specific surface area value. The pore size distribution of the ZHC was obtained from BJH method using the Density Functional Theory (DFT) method by ASiQwin software [8,9] (ASAP 2020, Micromeritics, Norcross, GA, USA).

### 2.4. Adsorption Process

The as-prepared ZHC can be applied as an efficient adsorbent for antibiotics in water. The antibiotics solution with increasing concentrations (TC, OFO 5–40 mg L^−1^, NFO 4–20 mg L^−1^, there chemical structures are shown in Appendix A) were carried out in a 100 mL conical flask with 5 mg ZHC; the adsorption begins when the solution is mixed with the absorbent. The adsorption experiments of three antibiotics were carried out at a constant speed (1000 rpm) in order to ensure the uniform diffusion of the absorbent in the solution. The different initial concentrations supernatant solution was obtained by filtering the solution with a syringe through a water filter membrane of 0.22 µm (SHZ-D (III) circulating water vacuum pump). After filtering, UV-vis spectra of the supernatant solution were recorded at a characteristic wavelength. A batch of adsorption experiments and blank experiments were conducted under constant temperature and pH value. The maximum adsorption amount of ZHC for TC, OFO and NFO is calculated using the formula (1):
(1)$$q_{e}=\frac{\left(C_{0}-C_{e}\right)V}{m}$$
where *q_e_* (mg/g) represents the absorption capacity of ZHC, *C*_0_ (mg/L) represents the initial concentration of TC/OFO/NFO, *C_e_* (mg/L) refers to the concentration of TC/OFO/NFO when the solution reaches the adsorption equilibrium, while *V* (L) represents the solution volume and *m* (g) represents the mass of the absorbent ZHC.

The adsorption of ZHC for antibiotics is based on Langmuir and Freundlich isothermal adsorption models. The Langmuir model formula [10,11] is as follows:
(2)$$\frac{C_{e}}{q_{e}}=\frac{1}{K_{L}q_{m}}+\frac{C_{e}}{q_{m}}$$
where *C_e_* is the equilibrium concentration, *K_L_* (L mg^−1^) is the adsorption constant, and *q_m_* (mg/g) is the maximum adsorption capacity.

The Freundlich model formula [7,12] is as follows:
(3)$$\ln q_{e}=\ln K_{f}+\left(\frac{1}{n}\right)\ln C_{e}$$
where *K_f_* ((mg g^−1^) × (L mg^−1^)^1/n^) and n represent the Freundlich constant. The higher the value, the better the adsorption effect.

The pseudo-first-order adsorption kinetics model of ZHC for antibiotics assumes that adsorption is controlled by diffusion steps and is given as [13]:
(4)$$\ln\left(q_{e}-q_{t}\right)=\ln q_{e}-K_{1}t$$
where *q_t_* (mg g^−1^) represents the amount of antibiotics absorbed in time *t* (min) and *K*_1_ (min^−1^) represents the first-order kinetic constant.

The pseudo-second-order kinetic model assumes that the adsorption rate is determined by the square value of the number of unoccupied adsorptive vacancies on the adsorbent surface, and the adsorption process is controlled by the chemical adsorption mechanism. The pseudo-second-order kinetic model [13,14] is as follows:
(5)$$\frac{t}{q_{t}}=\frac{1}{K_{2}{q_{e}}^{2}}+\frac{t}{q_{e}}$$
where *K*_2_ (g mg^−1^ min^−1^) represents the pseudo-second-order kinetic constant.

The formula of the intra-particle diffusion model [15,16] is as follows:
(6)qt=Kidt12+C
where *C* (mg g^−1^) is the constant involving the thickness, the boundary layer and *K_id_* (mg g^−1^ min^−1/2^) is the internal diffusion constant. In the particle diffusion model, *q_t_* and *t*^1/2^ are linearly fitted. If the straight line passes the origin, it is shown that the diffusion of the particles is the rate-limiting step of the adsorption process, and the adsorption process is controlled by the other adsorption stages if the origin is not passed.

## 3. Results and Discussion

To investigate the crystal structure of the obtained ZHC, XRD analysis was conducted. As shown in Figure 2a, the diffraction peaks (2θ) located at 26.61°, 43.45°, and 46.32° can be attributed to the (111), (100), and (110) crystal faces of graphite, respectively; which is in accordance with the peaks displayed in the graphite standard (JCPDS No. 01-075-2078). No impurity peak was found in the XRD pattern, demonstrating the high purity of the obtained ZHC materials. The morphology of ZIF-8 (Appendix A) and ZHC were explored by SEM. During the synthesis of ZIF-8, the addition of CTAB will significantly reduce the particle size of ZIF-8 crystals. It is speculated that CTAB may control the growth of crystals during synthesis, and the growth control mode may be as follows: Because of the anisotropic growth of ZIF-8 itself, part of CTAB is selectively adsorbed to the lowest level on the plane of mutual acting force, when CTAB is added into the precursor solution, CTAB in the process is more likely to act as a capping agent, because CTAB can be absorbed by the ZIF-8 crystals on the surface [17,18]. Therefore, the growth and separation of ZIF-8 crystals can be inhibited, and the morphology and size of ZIF-8 nanoparticles can be further controlled. According to the SEM images presented in Figure 2b,c, the particle size of the ZIF material witnessed a slight decrease after carbonization, which can be explained by the shrinkage of organic components during the calcination process. The TEM image displayed in Figure 2d confirms that the ZHC materials show obvious hollow structures.

N_2_ adsorption–desorption isotherms were carried out to investigate the pore diameter distribution and the specific surface area of ZHC. It can be seen from Figure 3a that the relative pressure (*P*/*P*_0_) between 0.2–0.9 shows a gradual upward trend of nitrogen adsorption and the pore diameter is concentrated at 5 nm (Figure 3b), which indicates that a mesoporous structure exists in the ZHC material. The adsorbent with a higher surface area (807.56 m^2^ g^−1^) can enhance the adsorption capacity of the adsorbent due to its abundant active adsorption sites. In addition, the mesoporous structure could give rise to the ability and efficiency of removing antibiotics pollutants [19].

According to the data in Table 1, the adsorption process of the ZHC on TC, NFO, and OFO is in accordance with the Langmuir (Figure 4a,b) and Freundlich (Figure 4c,d) models. However, considering the correlation fitting coefficient, the Langmuir model can describe the adsorption process well; that is, the adsorption of the material on TC, NFO, and OFO is homogeneous in the single molecular layer. The fitting of experimental data to model equations based on the Langmuir model shows the adsorption capacity of the ZHC on TC, NFO, and OFO are 267.3, 125.6, and 227.8 mg g^−1^, respectively.

As can be found in Table 2, the ZHC exhibits much higher antibiotics adsorption ability than pure ZIF-8 (Appendix A) and several inorganic materials in the reported literature, demonstrating the huge potential of the ZHC in adsorption applications.

Figure 5a shows that the adsorption of the ZHC to the three antibiotics reached equilibrium after ~48 h. Figure 5b,c are the fitted curves of pseudo-first-order dynamic models and pseudo-second-order dynamic models, respectively. Considering the correlation coefficient *R*^2^ values in Table 3, it can be concluded that the adsorption of the ZHC to the three antibiotics is more in line with the pseudo-second-order kinetic model, meaning the chemical adsorption mechanism exists in the adsorption process [7]. Figure 5d depicts the fitting curve of intra-particle diffusion model, which indicates that the adsorption process of the antibiotics can be divided into three stages: (1) antibiotics diffuse on the surface of the adsorbent; (2) antibiotics penetrate in the inner structure of the ZHC through pores; (3) antibiotics interact with the surface-active sites of the ZHC. The initial phase of stage (1) is regarded as the diffusion of antibiotics through the solution to the exterior surface of the ZHC, which can be known as external diffusion. Stage (2) shows the gradual adsorption where intraparticle diffusion is rate-limiting. Stage (3) is very fast and cannot be treated as the rate-controlling step, which is considered negligible. The regressions of *q_t_* vs. *t*^1/2^ for the antibiotic solutions were linear and did not pass through the origin, which means that the boundary layer diffusions were in control during the adsorption process and the intra-particle diffusions were involved in the adsorption process.

## 4. Conclusions

In summary, a novel ZIF-8-derived hollow carbon (ZHC) was developed as an efficient adsorbent to remove the residual antibiotics in the polluted water. The strong adsorption capacity was confirmed by Langmuir and Freundlich isothermal adsorption experiments, dynamic analysis, as well as theoretical calculation. The maximum quantities of ZHC on adsorbing tetracycline (TC), norfloxacin (NFO), and levofloxacin (OFO) are 267.3, 125.6, and 227.8 mg g^−^^1^, respectively. The excellent adsorptive performance of ZHC could be attributed to the hollow structure, which endows its high BET surface area and sufficient adsorption sites. Our results suggested that the MOF-derived carbon-material provides an extensible method to efficiently remove the antibiotics in the environment.

## Figures and Tables

**Figure 1 nanomaterials-09-00117-f001:**
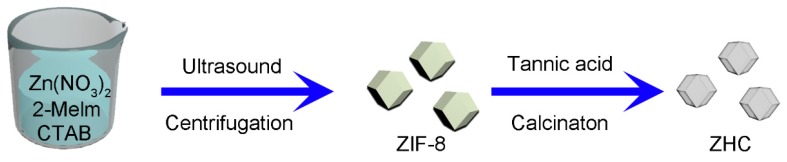
Schematic illustration of the fabrication process of ZIF-8 (zeolitic imidazolate framework)-derived hollow carbon (ZHC).

**Figure 2 nanomaterials-09-00117-f002:**
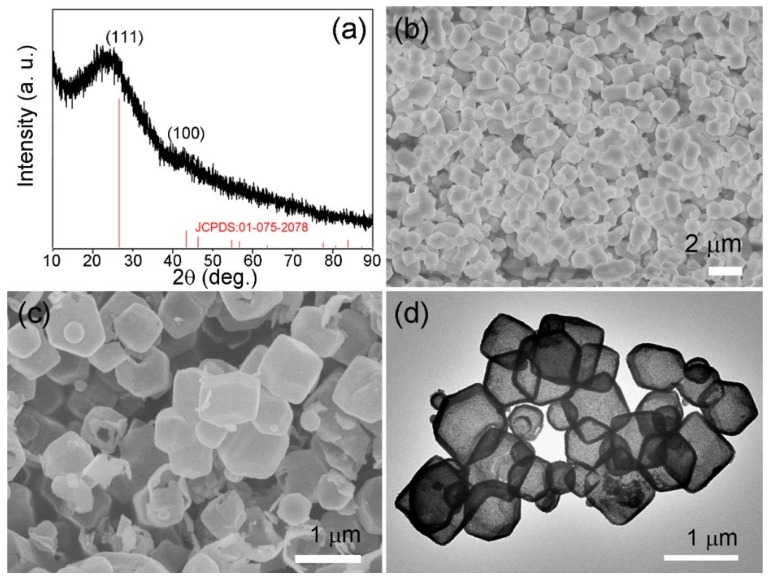
(**a**) X-ray diffraction (XRD) pattern of the ZHC; (**b**,**c**) scanning electron microscopy (SEM) image of ZIF-8 and ZHC; (**d**) transmission electron microscopy (TEM) image of the obtained ZHC.

**Figure 3 nanomaterials-09-00117-f003:**
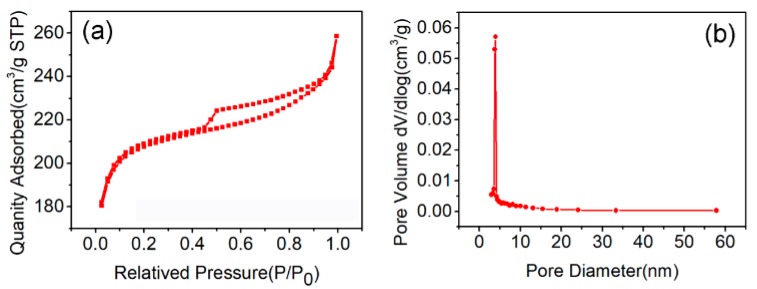
(**a**) N_2_ adsorption–desorption isotherm; and (**b**) pore size distribution of ZHC (N_2_ 77K, DFT).

**Figure 4 nanomaterials-09-00117-f004:**
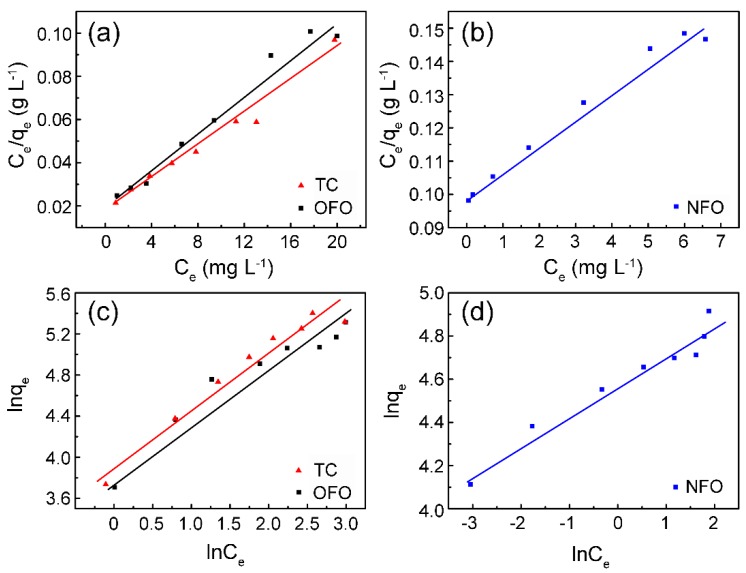
The isothermal adsorption fitting curves of Langmuir (**a**,**b**) and Freundlich (**c**,**d**) for tetracycline (TC), levofloxacin (OFO), and norfloxacin (NFO) by ZHC.

**Figure 5 nanomaterials-09-00117-f005:**
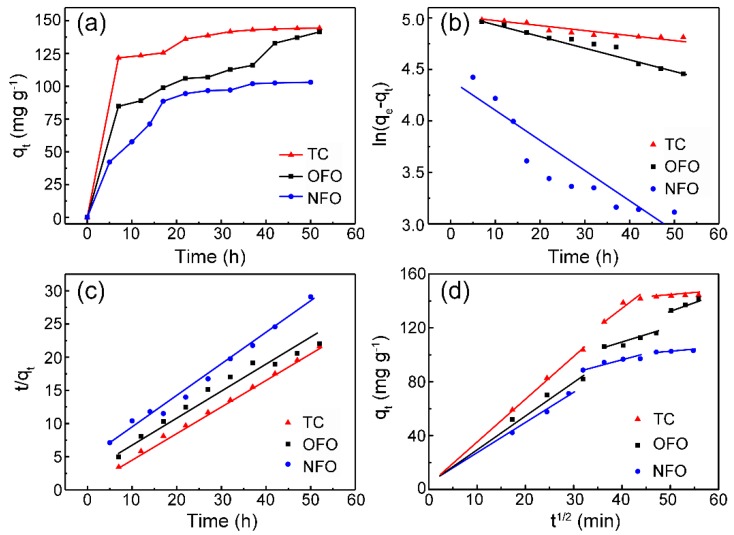
(**a**) Time curve fitting; (**b**) pseudo-first-order kinetic model; (**c**) pseudo-second-order kinetic model; and (**d**) the intra-particle diffusion model of ZHC.

**Table 1 nanomaterials-09-00117-t001:** The relevant fitting parameters of tetracycline (TC), norfloxacin (NFO), and levofloxacin (OFO) simulated using Langmuir and Freundlich models.

Pollutants	Langmuir			Freundlich		
	*K_L_* (L mg^−1^)	*q_m_* (mg g^−1^)	*R* ^2^	*K_f_* (mg g^−1^) (L mg^−1^)^1/n^	*n*	*R* ^2^
TC	0.212	267.3	0.972	70.80	1.8319	0.948
NFO	0.079	125.6	0.979	53.41	7.23	0.945
OFO	0.230	227.8	0.973	26.08	2.114	0.91

**Table 2 nanomaterials-09-00117-t002:** Comparison of the TC, NFO, and OFO adsorption ability of ZHC with other inorganic adsorbents.

Antibiotics	Adsorbents	*q_m_* (mg g^−1^)	Condition (pH)	Reference
Tetracycline	GN	2 × 10^−4^	7	[20]
	E_3_D_7_	133.3	8	[21]
	CNT-2%O	217.8	4	[22]
	Biochar	102	6	[23]
	ZIF-8	119.04	7	This work
	ZHC	267.3	7	This work
Norfloxacin	RGOS	50	6	[24]
	H-CNTS	76.3	7	[25]
	ZIF-8	38.69	7	This work
	ZHC	125.6	7	This work
Ofloxacin	GN	0.2	7	[20]
	BEPS-free biofilm-50	5.27	7	[26]
	Cassava residue-derived biochar	3.00	7	[27]
	ZIF-8	111.48	7	This work
	ZHC	227.8	7	This work

**Table 3 nanomaterials-09-00117-t003:** The related fitting parameters of the kinetic model.

Model	*C*_0_ (mg L^−1^)	Pseudo-First-Order			Pseudo-Second-Order		
		*K*_1_ (L min^−1^)	*q_e, cal_* (mg g^−1^)	*R* ^2^	*K*_2_ (g mg^−1^ min^−1^)	*q_e, cal_* (mg g^−1^)	*R* ^2^
TC	40	0.00422	148.21	0.852	4.33 × 10^−5^	151.89	0.998
NFO	20	0.02963	76.77	0.87	6.25 × 10^−5^	126.17	0.987
OFO	40	0.01136	158.92	0.954	3.83 × 10^−5^	161.60	0.961

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
