# Peer review of "ZIF-8-Derived Hollow Carbon for Efficient Adsorption of Antibiotics"

_nanomaterials, 2019, doi:10.3390/nano9010117_

Round 1

Reviewer 1 Report

The authors submitted a paper reporting the synthesis of a hollow carbon using ZIF-8 as sacrificial template, and testing the performance of the carbon material for the adsorption of 3 antibiotics (tetracycline, norfloxacin and levofloxacin). The hollow morphology of the carbon material obtained with the ZIF-8 sacrificial template is very interesting but a detailed evaluation of cost-to-income has to be addressed.

After a careful reading of the manuscript, the contribution of this work to the area of materials is clear and after major revision it may meet the novelty and impact criteria to be published in Nanomaterials journal.

In the next topics the revision comments are presented in order to correct some lacks and/or strengthen the manuscript:

1. At the end of the introduction section the author claim that “The work demonstrates the adsorptive performance of ZHC is comparable to the cutting-edge inorganic adsorbents and paves a new way for the design concept of effective low-cost adsorbents.” This is totally unacceptable considering the total cost of all the reagents and energy spent to produce de ZIF-8 derived hollow carbon. 

2. Regarding the use of MOFs as templates (introduction line 38): How does the cost of the derived materials (using MOFs as templates) compete with other porous and non-porous carbon materials (activated carbons, hydrochars, biochars, graphene, CNT…)? Does the performance of these carbon materials obtained after MOF sacrificial templating justify/covers their cost?

3. Introduction line 43, ref 4 is related to metal oxides not to carbons. Improve.

4. Ref 1 in the beginning of introduction must be a reference work, i.e. highly cited review, book and/or book chapter on this subject.

5. Clarify sentence “Since antibiotics cannot be assimilated by humans or animals, and it is inevitable to prevent the mass utilization of antibiotics, thus, a vicious cycle forms and result in the uncontrolled accumulation of residual antibiotics in the environment.” Antibiotics cannot be assimilated? If so how would they do “their job”?

6. The results regarding the antibiotics adsorption have to be fully revised and compared against ZIF-8 and other adsorbents.

7. Topic 2.3 lines 81/82: The BET equation does not allow to obtain the pore size distribution. Mention and describe the method used, add the corresponding reference.

8. Topic 2.4 Line 88: add the rpm used (constant speed).

9. Topic 2.4 lines 92-94: “A batch of adsorption experiments and blank experiments were conducted under constant temperature and pH values. The difference of pH value and temperature had an impact on the adsorption experiment.” According with the manuscript the authors did not evaluate the pH effect. Table 2 compared the data from the present study with literature but besides pH other experimental parameters could be distinct.

10. Line 97, qe definition is not correct

11. Line 102 a reference work must be cited [I. Langmuir, The adsorption of gases on plane surfaces of glass, mica and platinum, J. Am. Chem. Soc. 40 (1918) 1361–1403.]

12. Line 106 a reference work must be cited [H.M.F. Freundlich, Over the adsorption in solution, J. Phys. Chem. 57 (1906) 385–470.]

13. Lines 111 and 118 a reference work must be cited [Y.-S. Ho, Review of second-order models for adsorption systems, J. Hazard. Mater. 136 (2006) 681–689.]

14. Line 121 a reference work must be cited.

15. Lines 153 and 154 “In addition, the mesoporous structure always gives rise to the ability and efficiency of removing pollutants.” Which kind of pollutants? And the micropore volume?

16. Line 161: “simulation results”? or fitting of experimental data to model equations? Revise and correct accordingly?

17. The most common way of presenting liquid phase adsorption isotherms is to plot qe versus Ce using points to represent the experimental data (average values), error bars to illustrate the confidence level and lines to represent the fitting of the theoretical equations to the experimental data. Revise accordingly (Fig 4 can be presented in the main text of in supplementary)

18. Table 2 must also compare with activated carbons, largely commercialized and used in water treatment, and other carbon materials (biochars, hydrochars) if literature is available.

19. The most accurate way of presenting kinetic data is to plot C/C0 versus t since it allows to prove that no total removal of the pollutants occur. Correct accordingly.

20. Fig 5d, the authors use only 1 experimental point to define the first linear section for TC and OFO, this is not correct. After revising the just mentioned the discussion regarding the intra-particle must be improved since the manuscript only mentions the principles of this kinetic model.

21. Conclusion “Our results suggested that the MOF derived carbon-material provides an uncomplicated and extensible method to efficiently remove the antibiotics in the environment.” It is not acceptable to classify the procedure reported for the synthesis of the hollow carbons as uncomplicated when it requests the synthesis of a ZIF to be used as sacrificial template.

Author Response

# Reviewer 1

Thank you very much for your suggestions. Please find our reply in attachment.

Reviewer 2 Report

The authors studied ZIF-8 derived hollow carbon for efficient adsorption of three antibiotics, i.e. tetracycline, norfloxacin, and levofloxacin. It is reported that the MOF derived carbon-material provides extensible method to efficiently remove the above-mentioned antibiotics in the environment. I have checked the manuscript and I would like to address a significant/minor revision before it is accepted by Nanomaterials. In other words, after reading the draft work I have mixed feelings.

1) This work is very similar to the previous one (ref. 8, Li et al., J Mater Chem A, 2017, 5, 4352-4358) – see Fig. 2 – Figs. 2 and 3 [8], Fig. 3 – Fig. 3 [8], Fig. 4 – Fig. 4 [8], Fig. 5 – Fig. 5 [8], and tetracycline, norfloxacin, and levofloxacin - tetracycline, norfloxacin, and levofloxacin [8]. The similar results but for the different adsorbents. I would like to underline that I did not check other authors' work. It should be pointed out that there are infinitely many test materials/adsorbents for the investigations. What will be next?

2) page 2, lines 80-82. The Brunauer-80 Emmett-Teller (BET) method was used to provide the specific surface area value and the pore size 81 distribution of the ZHC (ASAP 2020, Micromeritics, America). A

I disagree with this statement. The BET equation has been developed to calculate only the surface area of a finely divided solid. This equation does neither provide a pore size nor a pore size distribution. Details about pore size distribution and the used method (Fig 3b) should be given.

3) The next problem is references – for example see eq. 2, i.e. “The Langmuir model formula” [8]. In ref. 8 [Li et al., J Mater Chem A, 2017, 5, 4352-4358] I found the following reference for this equation [J. Dai, J. He, A. Xie, L. Gao, J. Pan, X. Chen, Z. Zhou, X. Wei and Y. Yan, Chem. Eng. J., 2016, 284, 812]. The similar problem may be found for refs. 9 and 10 and the respective formulas. Homework for authors is to find the source articles.

4) The adsorption isotherm for TC, NFO, and OFO on the studied adsorbent should be collected in the same figure and added in the manuscript, for example, in the supplementary materials - similarly as for kinetic data in Fig. 5a. These results may be useful in the analysis of results and the estimation of the maximum adsorption of the studied organic compounds on ZIF-8 derived carbon. In my opinion they will be interested for readers

Author Response

The authors studied ZIF-8 derived hollow carbon for efficient adsorption of three antibiotics, i.e. tetracycline, norfloxacin, and levofloxacin. It is reported that the MOF derived carbon-material provides extensible method to efficiently remove the above-mentioned antibiotics in the environment. I have checked the manuscript and I would like to address a significant/minor revision before it is accepted by Nanomaterials. In other words, after reading the draft work I have mixed feelings.

Comment 1: This work is very similar to the previous one (ref. 8, Li et al., J Mater Chem A, 2017, 5, 4352-4358) – see Fig. 2 – Figs. 2 and 3 [8], Fig. 3 – Fig. 3 [8], Fig. 4 – Fig. 4 [8], Fig. 5 – Fig. 5 [8], and tetracycline, norfloxacin, and levofloxacin - tetracycline, norfloxacin, and levofloxacin [8]. The similar results but for the different adsorbents. I would like to underline that I did not check other authors' work. It should be pointed out that there are infinitely many test materials/adsorbents for the investigations. What will be next?

Response: Thanks for the comments. First of all, most adsorption processes of adsorbents are based on Langmuir and Freundlich adsorption models. Adsorption model formulas can be deformed, different formulas can get different graphs, including linear change graphs and non-linear change graphs. The reference and this paper’s work belong to graphs based on linear formulas, so the shapes are similar. TC is a tetracycline antibiotic, OFO and NFO are quinolone antibiotics, these three are typical representatives of antibiotics and are used as experimental subjects. We agree that the tested materials/adsorbents are infinitely and more efficient materials needs further research. The following work we are going to conduct is the heterogeneous porous carbon fiber materials and highly ordered transition metal oxides.

Comment 2: page 2, lines 80-82. The Brunauer-80 Emmett-Teller (BET) method was used to provide the specific surface area value and the pore size 81 distribution of the ZHC (ASAP 2020, Micromeritics, America).

I disagree with this statement. The BET equation has been developed to calculate only the surface area of a finely divided solid. This equation does neither provide a pore size nor a pore size distribution. Details about pore size distribution and the used method (Fig 3b) should be given.

Response: Thanks for the comments. The specific surface area and pore size distribution can be obtained by using the BET data measured via ASAP 2020 and BJH method using DFT method from ASiQwin software. We have added this part and highlighted in red in our revised manuscript.

Comment 3: The next problem is references – for example see eq. 2, i.e. “The Langmuir model formula” [8]. In ref. 8 [Li et al., J Mater Chem A, 2017, 5, 4352-4358] I found the following reference for this equation [J. Dai, J. He, A. Xie, L. Gao, J. Pan, X. Chen, Z. Zhou, X. Wei and Y. Yan, Chem. Eng. J., 2016, 284, 812]. The similar problem may be found for refs. 9 and 10 and the respective formulas. Homework for authors is to find the source articles.

Response: Thanks for your careful check. We have corrected them in the revised manuscript.

Comment 4: The adsorption isotherm for TC, NFO, and OFO on the studied adsorbent should be collected in the same figure and added in the manuscript, for example, in the supplementary materials - similarly as for kinetic data in Fig. 5a. These results may be useful in the analysis of results and the estimation of the maximum adsorption of the studied organic compounds on ZIF-8 derived carbon. In my opinion they will be interested for readers

Response: Thank you for your kind suggestion. The adsorption isotherm of NFO is not added with TC and OFO because of the initial concentration of NFO is different owing to its lower solubility compared with TC and NFO. We can’t collect them in the same figure.

Reviewer 3 Report

The authors present a interesting work base on modification of MOFs base material. Generally,  there are missing relevant facts related with the method of modification (by example how the temperature of calcinaton affects the sorption properties of the new material),basic chemical characterization of the material composition, how was the concentration of antibiotics in solution calculated?, surface area, functional groups on material surface,  Formula 1 is not correct (How do I  understand/accept the results if the basic formula is wrong?).  

Other remarks, 

In Line 42 MOF or MOFs

In Lines 49-51 described conclusions and should be aims.

Lines 90-92: “After  filtering, the supernatant solution for UV−visible spectroscopy was carried out at a characteristic  wavelength. …” What does it mean?

Line 93. “The difference of pH value and temperature had an impact on the adsorption experiment.” Conclusion in experimental part? If it is so which are the functional groups on the sorbent responsible of such impact?

Line 96: Formula 1 is not correct. How do I  understand/accept the results if the basic formula is wrong? 

Table 1: pollutions or pollutants?. Edition of table is wrong.

Explain conclusions from figure 5d (lines 178-181).

Line 175: “Considering the correlation coefficient R2 values”. R2 is determination coefficient.

Table 3: Which is the meaning of qe, cal? It was obtained from experimental values, isn´t it?

Lines 168-169: “…the ZHC exhibits much higher antibiotics adsorption ability than pure ZIF-8 (Fig. S3) and that of several inorganic materials in the reported literature, demonstrating the huge potential of the ZHC in adsorption applications…” How the authors explain the behavior of the materials for sorption of antibiotics presents in Figure S3 (a-d) as a function of the Ce?

How the authors explain the mechanisms of sorption of antibiotics on sorbent material? What´s about the conclusions from the results obtained from Langmuir and Freundlich models?

Author Response

Review 2:

Comment 1: The authors present an interesting work base on modification of MOFs base material. Generally, there are missing relevant facts related with the method of modification (by example how the temperature of calcination affects the sorption properties of the new material), basic chemical characterization of the material composition, how was the concentration of antibiotics in solution calculated?, surface area, functional groups on material surface,  Formula 1 is not correct (How do I understand/accept the results if the basic formula is wrong?).

Response: Thanks for the reviewer’s suggestion. The concentration of antibiotics is determined by Langer-Beer law. The unknown concentration of antibiotics can be calculated by standard curve and the absorbance of antibiotics measured. The specific surface area can be obtained by using the BET data measured by ASAP 2020. There are no functional groups on the surface of ZHC materials before use, because they are obtained at 800 oC, the functional groups cannot exist at such a high temperature. After adsorption, the functional groups are come from the antibiotics, such as -OH, -NH2 from TC, and etc. Formula 1 is a widely accepted and used formula. It was used in many reported influential literatures, for instance: Environ. Sci. Technol. 2017, 51, 12283.

Other remarks, 

Comment 2: In Line 42 MOF or MOFs

Response: Thanks for the comments. It is MOFs and we have corrected it.

Comment 3: In Lines 49-51 described conclusions and should be aims.

Response: Thanks for the comments. We have revised it accordingly.

Comment 4: Lines 90-92: “After filtering, the supernatant solution for UVvisible spectroscopy  was carried out at a characteristic wavelength. …” What does it mean?

Response: Thanks for the comments. It means the absorbance spectrum from UV-visible spectroscopy of TC/OFO/ NFO supernatant was measured at its specific characteristic wavelength. Each substance has its own specific characteristic wavelength.

Comment 5: Line 93. “The difference of pH value and temperature had an impact on the adsorption experiment.” Conclusion in experimental part? If it is so which are the functional groups on the sorbent responsible of such impact?

Response: Thanks for your suggestion, we have deleted this part in our revised manuscript for avoiding misunderstanding.

Comment 6: Line 96: Formula 1 is not correct. How do I understand/accept the results if the basic formula is wrong? 

Response: Thanks for your suggestions. Formula 1 is a widely used and accepted formula from the literature. It was used in many reported influential literatures, for instance: Environ. Sci. Technol. 2017, 51, 12283.

The above image is from Environ. Sci. Technol. 2017, 51, 12283, their work also uses Formula 1.

Comment 7: Table 1: pollutions or pollutants?. Edition of table is wrong.

Response: Thanks for the comments. It should be pollutants, we have corrected this error.

Comment 8: Explain conclusions from figure 5d (lines 178-181).

Response: Thanks for the comments. Fig. 5d corresponds to formula (5). In the intraparticle diffusion model, qt and t1/2 are linearly fitted, and the straight line passes through the origin. It shows that intraparticle diffusion is a rate-limiting step to control the adsorption process. The adsorption process is divided into three stages.

Comment 9: Line 175: “Considering the correlation coefficient R2 values”. R2 is determination coefficient.

Response: Thanks for the comments. We confirm that R2 is correlation coefficient.

Comment 10: Table 3: Which is the meaning of qe, cal? It was obtained from experimental values, isn´t it?

Response: Thanks for the comments. The qe, cal in Table 3 means empirical value, the maximum adsorption capacity was calculated by Langmuir, not obtained from experimental values.

Comment 11: Lines 168-169: “…the ZHC exhibits much higher antibiotics adsorption ability than pure ZIF-8 (Fig. S3) and that of several inorganic materials in the reported literature, demonstrating the huge potential of the ZHC in adsorption applications…” How the authors explain the behavior of the materials for sorption of antibiotics presents in Figure S3 (a-d) as a function of the Ce?

Response: Thanks for the comment. Fig. S3 describe the isothermal adsorption fitting curves of Langmuir (Fig S3a, 3b) and Freundlich (Fig S3c, S3d) for TC, OFO and NFO by pure ZIF-8. Langmuir calculation showed that the maximum adsorption capacity of pure ZIF-8 for TC, OFO and NFO was 119.04, 111.48 and 38.69 mg g-1, respectively. 

Comment 12: How the authors explain the mechanisms of sorption of antibiotics on sorbent material? What´s about the conclusions from the results obtained from Langmuir and Freundlich models ?

Response: Adsorption of antibiotics includes physical adsorption and chemical adsorption. The adsorption of antibiotics by adsorbents mainly depends on the specific surface area (porosity) of materials and the interaction between adsorbents and adsorbates. The pore of materials will precipitate some antibiotic molecules. The interaction forces between different adsorbent molecules and antibiotic molecules are different. The interaction forces include London dispersion force, dipole interaction, quadrupole interaction, electrostatic force and hydrogen bond, etc. They have different combined groups with antibiotic molecules to remove antibiotics.

Langmuir model is based on the uniformity of adsorbent surface, and there is no interaction between adsorbates. Adsorption is a monolayer adsorption, that is, the adsorption occurs only on the external surface of adsorbent. Freundlich adsorption equation is regarded as an adsorption isotherm on an inhomogeneous surface. Langmuir and Freundlich isothermal adsorption equations are classical adsorption models, which can calculate the maximum adsorption capacity of adsorbents and whether the adsorption is monolayer adsorption. The adsorption process in our work is monolayer adsorption owing to the higher R2 of Langmuir.

Round 2

Reviewer 1 Report

Please find the attached comments.

Author Response

Explanation of this revision

Manuscript ID: nanomaterials-411018

Comments to the Author

The authors submitted a revised version of the manuscript considering some of the comments enumerated in the first review process but still there were several issues not properly answered and reviewed. The revised version of the manuscript still needs extensive revision before considering its publication in Nanomaterial journal.

In the next topics the revision comments that must be properly answered are presented:

Previous comment 5.

Clarify sentence “Since antibiotics cannot be assimilated by humans or animals, and it is inevitable to prevent the mass utilization of antibiotics, thus, a vicious cycle forms and result in the uncontrolled accumulation of residual antibiotics in the environment.” Antibiotics cannot be assimilated? If so how would they do “their job”?

Authors Response: Thanks for your comments, we have been changed it to “Since antibiotics cannot be easily metabolized by humans or animals”.

Reviewer comment: It is more accurate to say that “antibiotics cannot be totally metabolized…”

Response: Thanks for the comments. We have changed it to “antibiotics cannot be totally metabolized…” in revised manuscript.

Previous comment 7. : Topic 2.3 lines 81/82: The BET equation does not allow to obtain the pore size distribution. Mention and describe the method used, add the corresponding reference.

Authors Response: Thanks for the comments. We have revised this part and added the reference according to your requirement, it can be seen in revised manuscript in red. And we have added the corresponding reference in revised manuscript:

8 Zhou, N.; Du, Y.; Wang, C.; Chen, R. Facile synthesis of hierarchically porous carbons by controlling the initial oxygen concentration in-situ carbonization of ZIF-8 for efficient water treatment, Chinese J Chem Eng, 2018, 10.1016/j.cjche.2018.05.014.

Reviewer comment:

- A reference to BET method must be added, the most recent IUPAC report is Thommes, M., Kaneko, K., Neimark, A.V., Olivier, J.P., Rodriguez-Reinoso, F., Rouquerol, J., Sing, K.S.W. 2015. Physisorption of gases, with special reference to the evaluation of surface area and pore size distribution (IUPAC Technical Report). Pure Appl. Chem., 87(9-10), 1051-1069.

- A reference regarding the BJH method itself must also be added. The one the authors cited is probably a work that applies the method and not a reference work, thus is not an acceptable reference.

Response: Thanks for the comments, We have changed ref. 8 according to your suggestion.

8. Thommes, M.; Kaneko, K.; Neimark, A.V.; Olivier, J.P.; Rodriguez-Reinoso, F.; Rouquerol, J.; Sing, K.S.W. Physisorption of gases, with special reference to the evaluation of surface area and pore size distribution (IUPAC Technical Report). Pure Appl. Chem., 2015, 87(9-10), 1051-1069.

As regarding the BJH method, we have added a reference as follow:

9. Francisco, J. S.; Katie, A. C.; Matthias T. Characterization of Micro/Mesoporous Materials by Physisorption: Concepts and Case Studies. Acc. Mater. Surf. Res., 2018, 3(2), 34-50.

Previous comment 9:Topic 2.4 lines 92-94: “A batch of adsorption experiments and blank experiments were conducted under constant temperature and pH values. The difference of pH value and temperature had an impact on the adsorption experiment.” According with the manuscript the authors did not evaluate the pH effect. Table 2 compared the data from the present study with literature but besides pH other experimental parameters could be distinct.

Authors Response: Thanks for the comments. Because the antibiotic solution in this work is based on deionized water as solvent, the pH value is 7, and we didn’t evaluate the effect of PH value on adsorption. In revised manuscript, we have deleted the sentence of “The difference of pH value and temperature had an impact on the adsorption experiment.” From the literature, it is hard to find so many works just carried out at pH= 7.

Reviewer comment: The authors answer is not acceptable! It is not true that the use of deionized water as solvent for the preparations of antibiotic solutions gives rise to solutions with pH 7, it depends on the antibiotic used and the pH of the prepared solutions must always be checked.

Response: Thanks for the comments. We have checked the pH value of the antibiotic solution by pH indicator paper, the color of indicator paper showed the value is about 7, the antibiotic solutions are of very low concentration, so the pH values have no obvious changes.

Previous comment 14. Line 121 a reference work must be cited.

Authors Response: A related reference work has been cited as follow:

15 Liu, Q.; Zhong, L.; Zhao, Q.; Frear, C.; Zheng Y. Synthesis of Fe3O4/Polyacrylonitrile Composite Electrospun Nanofiber Mat for Effective Adsorption of Tetracycline, ACS Appl. Mater. Interfaces, 2015, 7, 14573.

Reviewer comment: The added reference is not a work related to the intra-particle diffusion models, a reference work must be cited.

Response: Thanks for the comments. We can not agree with the reviewer about this comment, this reference is a work related to the intra-particle diffusion models, the intra-particle diffusion model is also named with Weber-Morris kinetics modeling in the reference, the calculation fomula of the intra-particle diffusion model and the Weber-Morris kinetics modeling is the same, the original information in the reference is as follow:

Previous comment 15:Lines 153 and 154 “In addition, the mesoporous structure always gives rise to the ability and efficiency of removing pollutants.” Which kind of pollutants? And the micropore volume?

Authors Response: Thanks for the comments, here the pollutants refer to the antibiotics. We have revised this sentence to “In addition, the mesoporous structure always gives rise to the ability and efficiency of removing antibiotics pollutants”. What we prepared is mesoporous structure materials, so there are no data for the micropore volume.

Reviewer comment: Literature data do not always prove that the mesopore volume “always gives rise to the ability and efficiency of removing pollutants” in specific antibiotics. This affirmation is not correct and thus must be revised since it always depends on de pore size distributions, on the pollutant and on the experimental conditions of the assay. To say that there are literature studies that demonstrate that the mesoporous structure improved the ability and efficiency of an adsorbent materials for removal pollutants, namely antibiotics, the authors must cite specific works where this is proved.

Response: Thanks for the comments. The word “always” is not rigorous here, so we have changed this sentence to “In addition, the mesoporous structure could give rise to the ability and efficiency of removing antibiotics pollutants.19”, and we have added a related reference as follow:

19. Ahmad, Z. U.; Yao, L.; Wang, J.; Gang, D. D.; Islam, F.; Lian, Q.; Zappi, M. E.. Neodymium embedded ordered mesoporous carbon (OMC) for enhanced adsorption of sunset yellow: Characterizations, adsorption study and adsorption mechanism. Chem. Eng. J., 2019, 359, 814-826.

Previous comment 17: The most common way of presenting liquid phase adsorption isotherms is to plot qe versus Ce using points to represent the experimental data (average values), error bars to illustrate the confidence level and lines to represent the fitting of the theoretical equations to the experimental data. Revise accordingly (Fig 4 can be presented in the main text of in supplementary)

Authors Response: Thanks for the comments, we agree that qe vs Ce is a common way of presenting liquid phase adsorption isotherms, while the Ce/qe vs Ce is also an acceptable and right way to present liquid phase adsorption isotherms. In this case, in the current work, the error bars are not needed, we use R2 to express the relationship between experimental data and the theoretical formulas.

Reviewer comment: The R2 value represents the quality of the linear fitting between the experimental data and the theoretical equations. Thus the R2 value does not give any information regarding the range of values obtained during the experiment. The kinetic and equilibrium experimental data acquisition must be performed in replicas, and so in the graphics the mean value is usually presented by a symbol and the error bars illustrate the range of values obtained for each experimental point. The answer of the authors is not acceptable and the manuscript has to be revised.

Response: Thanks for the comments. We can not agree with the reviewer about this comment. In our work, the Ce/qe vs. Ce is also an identified, acceptable and right way to present liquid phase adsorption isotherms. In this case, it is not necessary to use qe versus Ce as requsted by the Referee. In addition, the error bars is not a necessity for these plots. Similar work in reported literatures are shown as follow, error bars are not exist:

Fig. 1. The Langmuir isotherm model isothermal fittings for TET adsorption on Fe/Zn-biochar. (Bioresour Technol. 2017, 245, 266-273.)

Figure 2. The equilibrium isotherms for TC adsorbed by CLDHs/BC of the Langmuir model (Sci Rep, 2016, 6, 39691)

Previous comment 19:The most accurate way of presenting kinetic data is to plot C/C0 versus t since it allows to prove that no total removal of the pollutants occur. Correct accordingly.

Authors Response: Thanks for the comments. C/C0 vs t is based on non-linear mathematical formulas, this work is based on linear mathematical formulas. In fact, there are no essential difference between the results and parameters obtained by non-linear and linear fitting, it is only a single deformation of the mathematical formulas. The result is accurate and reliable.

Reviewer comment: The graphic in figure 5(a) of the revised version is a non-linear representation of kinetic data so the authors’ response does not answer to the original comment. The most accurate way of presenting kinetic data is to plot C/C0 versus t since it allows to prove that no total removal of the pollutants occur. If the authors do not correct the figure accordingly in the revised version of the manuscript at least the authors have to prove that in the plateau no total removal of the antibiotic occur (Calculate Ct at the plateau).

Response:                Thanks for the comments. In this paper, antibiotics were not completely removed when the adsorption reached equilibrium. Ct represents the amount of adsorption at any time in the adsorption process. Ce refers to the amount of adsorption at equilibrium. When the adsorption reaches equilibrium, Ct is Ce.

The adsorption amount of completely removed antibiotics could be caculated from formula 1:

Taking TC as an example, the initial amount of C0 is 40 mg/L, Ce is 0 when antibiotics are completely removed, antibiotics solution volume (V) is 50 mL, adsorbent mass (m) is 5 mg. It can be caculated that TC can be completely removed only when the adsorption capacity of TC was 400 mg/g, while our result is 267.3 mg/g.

Previous comment 20:Fig 5d, the authors use only 1 experimental point to define the first linear section for TC and OFO, this is not correct. After revising the just mentioned the discussion regarding the intra-particle must be improved since the manuscript only mentions the principles of this kinetic model.

Authors Response: Thanks for the comments. The problem you mentioned has been corrected by a supplementary experiment. The revised Fig 5d is shown as follow…

Reviewer comment: Although the authors performed more experimental data to support the fitting of the intra-particle model the discussion regarding the fitting of this model is still absence since authors only enumerate the principles of the model. It has to be improved.

Response: Thanks for the comments. We have added the discussion and changed this part as follow:

Fig. 5d depicts the fitting curve of intra-particle diffusion model, which indicates that the adsorption process of antibiotics can be divided into three stages: (1) antibiotics diffuse on the surface of the adsorbent, (2) antibiotics penetrate in the inner structure of ZHC through pores, (3) antibiotics interact with the surface-active sites of ZHC. The initial phase of stage (1) is regarded as the diffusion of antibiotics through the solution to the exterior surface of the ZHC, which can be known as external diffusion. Stage (2) shows the gradual adsorption where intraparticle diffusion is rate-limiting. Stage (3) is very fast and cannot be treated as the rate-controlling step, which is considered negligible. The regressions of qt vs. t1/2 for the antibiotic solutions were linear and did not pass through the origin, which means that the boundary layer diffusions were in control during the adsorption process and the intra-particle diffusions were involved in the adsorption process.

Reviewer 2 Report

I am pleased to see an emerging improvement. The authors have satisfactorily responded to all my questions and made the necessary changes to the manuscript.  In my opinion, no further reviewing is needed. I leave  the final decision to the second reviewer because of the multiplicity of comments.

Author Response

Thank you.

Reviewer 3 Report

In brief:

line 92: "...the supernatant solution for UV−visible spectra was recorded at..." If I guess the authors would like to say UV-vis spectra of the supernatant solution was recorded. Perhaps, they didn't know the difference or they overlooked it.

Formula 1: results are expressed as mg/g, I can say that the formula is not correct. According to formula 1 in manuscript: (mg*g/L)/V. Authors didn´t express units for V (L, mL)????.

lines 42 to 52: "The maximum adsorption capacities 49 of ZHC for TC, NFO, and OFO are 267.3, 125.6, and 227.8 mg g-1, respectively, according to the data obtained from Langmuir and Freundlich isothermal adsorption simulation as well as kinetic experiments. The adsorptive performance of ZHC is comparable to the cutting-edge inorganic adsorbents and paves a new way for the design concept of adsorbents." If are they conclusions from their work why they included in the Introduction?

Line 176: "Considering the correlation coefficient R2 values..." the authors ignore that R2 denotes coefficient of determination and not correlation coefficient. 

Author Response

Comment 1: line 92: "...the supernatant solution for UVvisible spectra was recorded at..." If I guess the authors would like to say UV-vis spectra of the supernatant solution was recorded. Perhaps, they didn't know the difference or they overlooked it.

Response: Thank you for the comments. according to your suggestion, we have changed this sentence to ‘After filtering, UV-vis spectra of the supernatant solution were recorded at a characteristic wavelength.’

Comment 2: Formula 1: results are expressed as mg/g, I can say that the formula is not correct. According to formula 1 in manuscript: (mg*g/L)/V. Authors didn´t express units for V (L, mL)????.

Response: Thank you for your carefully check. the Formula 1 in our manuscript is not correct, the position of V and m are reversed, we have corrected this writing error as follow:

In addition, we added the units of the relevant parameters in the formula as follows:

where qe (mg/g) represents the absorption capacity of ZHC, C0 (mg/L) represents the initial concentration of TC/OFO/NFO, Ce (mg/L) refers to the concentration of TC/OFO/NFO when the solution reaches the adsorption equilibrium, while V (L) represents the solution volume and m (g) represents the mass of the absorbent ZHC.

Finally, although the formula 1 is incorrectly written, we calculate the data according to the correct formula. We sincerely thank you for the careful check and comment.

Comment 3: lines 42 to 52: "The maximum adsorption capacities 49 of ZHC for TC, NFO, and OFO are 267.3, 125.6, and 227.8 mg g-1, respectively, according to the data obtained from Langmuir and Freundlich isothermal adsorption simulation as well as kinetic experiments. The adsorptive performance of ZHC is comparable to the cutting-edge inorganic adsorbents and paves a new way for the design concept of adsorbents." If are they conclusions from their work why they included in the Introduction?

Response: We have changed these sentences to “The maximum adsorption capacities of ZHC for TC, NFO, and OFO are investigated according to the data obtained from Langmuir and Freundlich isothermal adsorption simulation as well as kinetic experiments.

Comment 4: Line 176: "Considering the correlation coefficient R2 values..." the authors ignore that R2 denotes coefficient of determination and not correlation coefficient.

Response: Thank you for the comment. Usually, R (not R2) denotes to coefficient of determination. In this paper, here we can confirm that the R2 represents the correlation coefficient, the relevant references are as follows:

Z, Li. X, Meng. Z, Zhang, Mater. Res. Bull. 2019, 111, 238-244.

Zaki Uddin AhmadL, YaoJ, WangChem. Eng. J. 2019, 359, 814-826.